# Learning to learn by gradient descent
# by gradient descent

**Marcin Andrychowicz[1], Misha Denil[1], Sergio Gómez Colmenarejo[1], Matthew W. Hoffman[1],
David Pfau[1], Tom Schaul[1], Brendan Shillingford[1,2], Nando de Freitas[1,2,3]**

[1]Google DeepMind    [2]University of Oxford    [3]Canadian Institute for Advanced Research

marcin.andrychowicz@gmail.com
{mdenil,sergomez,mwhoffman,pfau,schaul}@google.com
brendan.shillingford@cs.ox.ac.uk, nandodefreitas@google.com

## Abstract

The move from hand-designed features to learned features in machine learning has
been wildly successful. In spite of this, optimization algorithms are still designed
by hand. In this paper we show how the design of an optimization algorithm can be
cast as a learning problem, allowing the algorithm to learn to exploit structure in
the problems of interest in an automatic way. Our learned algorithms, implemented
by LSTMs, outperform generic, hand-designed competitors on the tasks for which
they are trained, and also generalize well to new tasks with similar structure. We
demonstrate this on a number of tasks, including simple convex problems, training
neural networks, and styling images with neural art.

## 1   Introduction

Frequently, tasks in machine learning can be expressed as the problem of optimizing an objective
function $f(\theta)$ defined over some domain $\theta \in \Theta$. The goal in this case is to find the minimizer
$\theta^* = \arg\min_{\theta \in \Theta} f(\theta)$. While any method capable of minimizing this objective function can be
applied, the standard approach for differentiable functions is some form of gradient descent, resulting
in a sequence of updates

$$\theta_{t+1} = \theta_t - \alpha_t \nabla f(\theta_t).$$

The performance of vanilla gradient descent, however, is hampered by the fact that it *only* makes use
of gradients and ignores second-order information. Classical optimization techniques correct this
behavior by rescaling the gradient step using curvature information, typically via the Hessian matrix
of second-order partial derivatives—although other choices such as the generalized Gauss-Newton
matrix or Fisher information matrix are possible.

Much of the modern work in optimization is based around designing update rules tailored to specific
classes of problems, with the types of problems of interest differing between different research
communities. For example, in the deep learning community we have seen a proliferation of optimiza-
tion methods specialized for high-dimensional, non-convex optimization problems. These include
momentum [Nesterov, 1983, Tseng, 1998], Rprop [Riedmiller and Braun, 1993], Adagrad [Duchi
et al., 2011], RMSprop [Tieleman and Hinton, 2012], and ADAM [Kingma and Ba, 2015]. More
focused methods can also be applied when more structure of the optimization problem is known
[Martens and Grosse, 2015]. In contrast, communities who focus on sparsity tend to favor very
different approaches [Donoho, 2006, Bach et al., 2012]. This is even more the case for combinatorial
optimization for which relaxations are often the norm [Nemhauser and Wolsey, 1988].

This industry of optimizer design allows different communities to create optimization methods which exploit structure in their problems of interest at the expense of potentially poor performance on problems outside of that scope. Moreover the *No Free Lunch Theorems for Optimization* [Wolpert and Macready, 1997] show that in the setting of combinatorial optimization, no algorithm is able to do better than a random strategy in expectation. This suggests that specialization to a subclass of problems is in fact the *only* way that improved performance can be achieved in general.

In this work we take a different tack and instead propose to replace hand-designed update rules with a learned update rule, which we call the optimizer $g$, specified by its own set of parameters $\phi$. This results in updates to the optimizee $f$ of the form

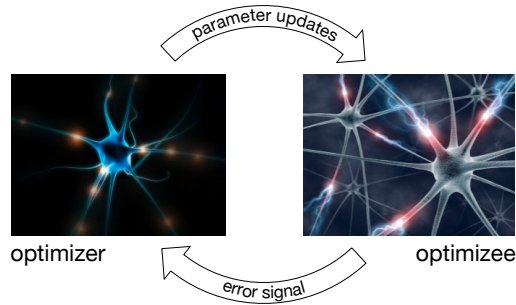

Figure 1: The optimizer (left) is provided with performance of the optimizee (right) and proposes updates to increase the optimizee's performance. [photos: Bobolas, 2009, Maley, 2011]

$$\theta_{t+1} = \theta_t + g_t(\nabla f(\theta_t), \phi). \tag{1}$$

A high level view of this process is shown in Figure 1. In what follows we will explicitly model the update rule $g$ using a recurrent neural network (RNN) which maintains its own state and hence dynamically updates as a function of its iterates.

## 1.1 Transfer learning and generalization

The goal of this work is to develop a procedure for constructing a learning algorithm which performs well on a particular class of optimization problems. Casting algorithm design as a learning problem allows us to specify the class of problems we are interested in through example problem instances. This is in contrast to the ordinary approach of characterizing properties of interesting problems analytically and using these analytical insights to design learning algorithms by hand.

It is informative to consider the meaning of *generalization* in this framework. In ordinary statistical learning we have a particular function of interest, whose behavior is constrained through a data set of example function evaluations. In choosing a model we specify a set of inductive biases about how we think the function of interest should behave at points we have not observed, and generalization corresponds to the capacity to make predictions about the behavior of the target function at novel points. In our setting the examples are themselves *problem instances*, which means generalization corresponds to the ability to transfer knowledge between different problems. This reuse of problem structure is commonly known as *transfer learning*, and is often treated as a subject in its own right. However, by taking a meta-learning perspective, we can cast the problem of transfer learning as one of generalization, which is much better studied in the machine learning community.

One of the great success stories of deep-learning is that we can rely on the ability of deep networks to generalize to new examples by learning interesting sub-structures. In this work we aim to leverage this generalization power, but also to lift it from simple supervised learning to the more general setting of optimization.

## 1.2 A brief history and related work

The idea of using *learning to learn* or *meta-learning* to acquire knowledge or inductive biases has a long history [Thrun and Pratt, 1998]. More recently, Lake et al. [2016] have argued forcefully for its importance as a building block in artificial intelligence. Similarly, Santoro et al. [2016] frame multi-task learning as generalization, however unlike our approach they directly train a base learner rather than a training algorithm. In general these ideas involve learning which occurs at two different time scales: rapid learning within tasks and more gradual, *meta* learning across many different tasks.

Perhaps the most general approach to meta-learning is that of Schmidhuber [1992, 1993]—building on work from [Schmidhuber, 1987]—which considers networks that are able to modify their own weights. Such a system is differentiable end-to-end, allowing both the network and the learning

algorithm to be trained jointly by gradient descent with few restrictions. However this generality comes at the expense of making the learning rules very difficult to train. Alternatively, the work of Schmidhuber et al. [1997] uses the Success Story Algorithm to modify its search strategy rather than gradient descent; a similar approach has been recently taken in Daniel et al. [2016] which uses reinforcement learning to train a controller for selecting step-sizes.

Bengio et al. [1990, 1995] propose to learn updates which avoid back-propagation by using simple parametric rules. In relation to the focus of this paper the work of Bengio et al. could be characterized as *learning to learn **without** gradient descent by gradient descent*. The work of Runarsson and Jonsson [2000] builds upon this work by replacing the simple rule with a neural network.

Cotter and Conwell [1990], and later Younger et al. [1999], also show fixed-weight recurrent neural networks can exhibit dynamic behavior without need to modify their network weights. Similarly this has been shown in a filtering context [e.g. Feldkamp and Puskorius, 1998], which is directly related to simple multi-timescale optimizers [Sutton, 1992, Schraudolph, 1999].

Finally, the work of Younger et al. [2001] and Hochreiter et al. [2001] connects these different threads of research by allowing for the output of backpropagation from one network to feed into an additional *learning* network, with both networks trained jointly. Our approach to meta-learning builds on this work by modifying the network architecture of the optimizer in order to scale this approach to larger neural-network optimization problems.

## 2 Learning to learn with recurrent neural networks

In this work we consider directly parameterizing the optimizer. As a result, in a slight abuse of notation we will write the final *optimizee parameters* $\theta^*(f, \phi)$ as a function of the *optimizer parameters* $\phi$ and the function in question. We can then ask the question: What does it mean for an optimizer to be good? Given a distribution of functions $f$ we will write the expected loss as

$$\mathcal{L}(\phi) = \mathbb{E}_f \Big[ f\big(\theta^*(f, \phi)\big) \Big] . \tag{2}$$

As noted earlier, we will take the update steps $g_t$ to be the output of a recurrent neural network $m$, parameterized by $\phi$, whose state we will denote explicitly with $h_t$. Next, while the objective function in (2) depends only on the final parameter value, for training the optimizer it will be convenient to have an objective that depends on the entire trajectory of optimization, for some horizon T,

$$\mathcal{L}(\phi) = \mathbb{E}_f \left[ \sum_{t=1}^{T} w_t f(\theta_t) \right] \qquad \text{where} \qquad \theta_{t+1} = \theta_t + g_t , \tag{3}$$
$$\begin{bmatrix} g_t \\ h_{t+1} \end{bmatrix} = m(\nabla_t, h_t, \phi) .$$

Here $w_t \in \mathbb{R}_{\geq 0}$ are arbitrary weights associated with each time-step and we will also use the notation $\nabla_t = \nabla_\theta f(\theta_t)$. This formulation is equivalent to (2) when $w_t = \mathbf{1}[t = T]$, but later we will describe why using different weights can prove useful.

We can minimize the value of $\mathcal{L}(\phi)$ using gradient descent on $\phi$. The gradient estimate $\partial \mathcal{L}(\phi)/\partial \phi$ can be computed by sampling a random function $f$ and applying backpropagation to the computational graph in Figure 2. We allow gradients to flow along the solid edges in the graph, but gradients along the dashed edges are dropped. Ignoring gradients along the dashed edges amounts to making the assumption that the gradients of the optimizee do not depend on the optimizer parameters, i.e. $\partial \nabla_t/\partial \phi = 0$. This assumption allows us to avoid computing second derivatives of $f$.

Examining the objective in (3) we see that the gradient is non-zero only for terms where $w_t \neq 0$. If we use $w_t = \mathbf{1}[t = T]$ to match the original problem, then gradients of trajectory prefixes are zero and only the final optimization step provides information for training the optimizer. This renders Backpropagation Through Time (BPTT) inefficient. We solve this problem by relaxing the objective such that $w_t > 0$ at intermediate points along the trajectory. This changes the objective function, but allows us to train the optimizer on partial trajectories. For simplicity, in all our experiments we use $w_t = 1$ for every $t$.

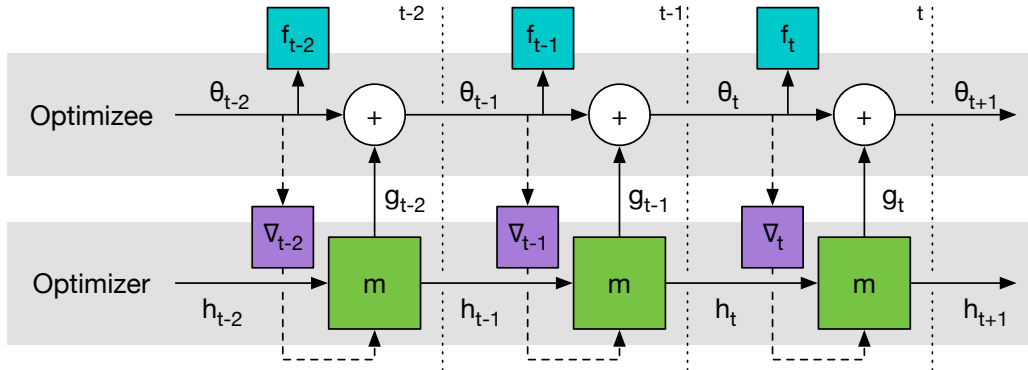

Figure 2: Computational graph used for computing the gradient of the optimizer.

## 2.1 Coordinatewise LSTM optimizer

One challenge in applying RNNs in our setting is that we want to be able to optimize at least tens of thousands of parameters. Optimizing at this scale with a fully connected RNN is not feasible as it would require a huge hidden state and an enormous number of parameters. To avoid this difficulty we will use an optimizer $m$ which operates coordinatewise on the parameters of the objective function, similar to other common update rules like RMSprop and ADAM. This *coordinatewise network architecture* allows us to use a very small network that only looks at a single coordinate to define the optimizer and share optimizer parameters across different parameters of the optimizee.

Different behavior on each coordinate is achieved by using separate activations for each objective function parameter. In addition to allowing us to use a small network for this optimizer, this setup has the nice effect of making the optimizer invariant to the order of parameters in the network, since the same update rule is used independently on each coordinate.

We implement the update rule for each coordinate using a two-layer Long Short Term Memory (LSTM) network [Hochreiter and Schmidhuber, 1997], using the now-standard forget gate architecture. The network takes as input the optimizee gradient for a single coordinate as well as the previous hidden state and outputs the update for the corresponding optimizee parameter. We will refer to this architecture, illustrated in Figure 3, as an LSTM optimizer.



Figure 3: One step of an LSTM optimizer. All LSTMs have shared parameters, but separate hidden states.

The use of recurrence allows the LSTM to learn dynamic update rules which integrate information from the history of gradients, similar to momentum. This is known to have many desirable properties in convex optimization [see e.g. Nesterov, 1983] and in fact many recent learning procedures—such as ADAM—use momentum in their updates.

**Preprocessing and postprocessing** Optimizer inputs and outputs can have very different magnitudes depending on the class of function being optimized, but neural networks usually work robustly only for inputs and outputs which are neither very small nor very large. In practice rescaling inputs and outputs of an LSTM optimizer using suitable constants (shared across all timesteps and functions $f$) is sufficient to avoid this problem. In Appendix A we propose a different method of preprocessing inputs to the optimizer inputs which is more robust and gives slightly better performance.

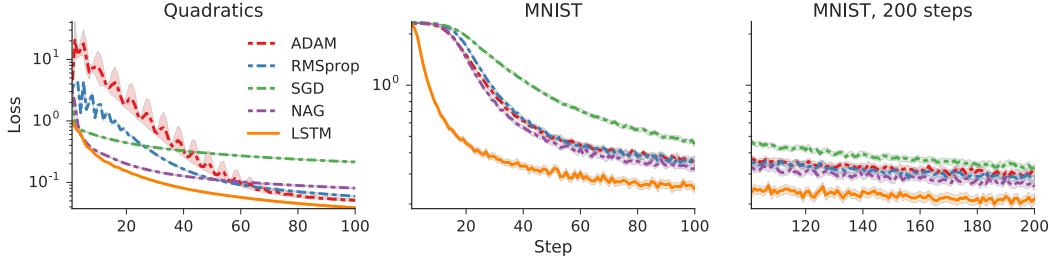

Figure 4: Comparisons between learned and hand-crafted optimizers performance. Learned optimizers are shown with solid lines and hand-crafted optimizers are shown with dashed lines. Units for the $y$ axis in the MNIST plots are logits. **Left:** Performance of different optimizers on randomly sampled 10-dimensional quadratic functions. **Center:** the LSTM optimizer outperforms standard methods training the base network on MNIST. **Right:** Learning curves for steps 100-200 by an optimizer trained to optimize for 100 steps (continuation of center plot).

# 3    Experiments

In all experiments the trained optimizers use two-layer LSTMs with 20 hidden units in each layer. Each optimizer is trained by minimizing Equation 3 using truncated BPTT as described in Section 2. The minimization is performed using ADAM with a learning rate chosen by random search.

We use early stopping when training the optimizer in order to avoid overfitting the optimizer. After each epoch (some fixed number of learning steps) we freeze the optimizer parameters and evaluate its performance. We pick the best optimizer (according to the final validation loss) and report its average performance on a number of freshly sampled test problems.

We compare our trained optimizers with standard optimizers used in Deep Learning: SGD, RMSprop, ADAM, and Nesterov's accelerated gradient (NAG). For each of these optimizer and each problem we tuned the learning rate, and report results with the rate that gives the best final error for each problem. When an optimizer has more parameters than just a learning rate (e.g. decay coefficients for ADAM) we use the default values from the `optim` package in Torch7. Initial values of all optimizee parameters were sampled from an IID Gaussian distribution.

## 3.1    Quadratic functions

In this experiment we consider training an optimizer on a simple class of synthetic 10-dimensional quadratic functions. In particular we consider minimizing functions of the form

$$f(\theta) = \|W\theta - y\|_2^2$$

for different 10x10 matrices $W$ and 10-dimensional vectors $y$ whose elements are drawn from an IID Gaussian distribution. Optimizers were trained by optimizing random functions from this family and tested on newly sampled functions from the same distribution. Each function was optimized for 100 steps and the trained optimizers were unrolled for 20 steps. We have not used any preprocessing, nor postprocessing.

Learning curves for different optimizers, averaged over many functions, are shown in the left plot of Figure 4. Each curve corresponds to the average performance of one optimization algorithm on many test functions; the solid curve shows the learned optimizer performance and dashed curves show the performance of the standard baseline optimizers. It is clear the learned optimizers substantially outperform the baselines in this setting.

## 3.2    Training a small neural network on MNIST

In this experiment we test whether trainable optimizers can learn to optimize a small neural network on MNIST, and also explore how the trained optimizers generalize to functions beyond those they were trained on. To this end, we train the optimizer to optimize a base network and explore a series of modifications to the network architecture and training procedure at test time.

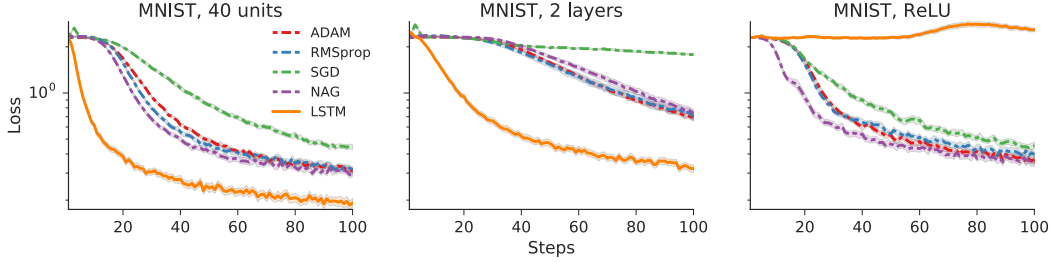

Figure 5: Comparisons between learned and hand-crafted optimizers performance. Units for the $y$ axis are logits. **Left:** Generalization to the different number of hidden units (40 instead of 20). **Center:** Generalization to the different number of hidden layers (2 instead of 1). This optimization problem is very hard, because the hidden layers are very narrow. **Right:** Training curves for an MLP with 20 hidden units using ReLU activations. The LSTM optimizer was trained on an MLP with sigmoid activations.

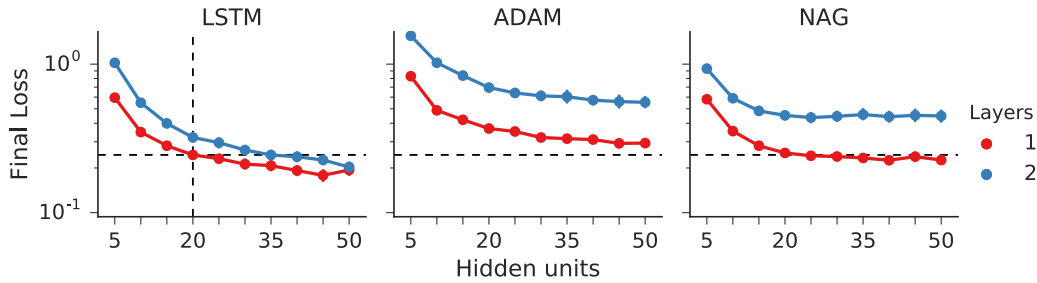

Figure 6: Systematic study of final MNIST performance as the optimizee architecture is varied, using sigmoid non-linearities. The vertical dashed line in the left-most plot denotes the architecture at which the LSTM is trained and the horizontal line shows the final performance of the trained optimizer in this setting.

In this setting the objective function $f(\theta)$ is the cross entropy of a small MLP with parameters $\theta$. The values of $f$ as well as the gradients $\partial f(\theta)/\partial\theta$ are estimated using random minibatches of 128 examples. The base network is an MLP with one hidden layer of 20 units using a sigmoid activation function. The only source of variability between different runs is the initial value $\theta_0$ and randomness in minibatch selection. Each optimization was run for 100 steps and the trained optimizers were unrolled for 20 steps. We used input preprocessing described in Appendix A and rescaled the outputs of the LSTM by the factor $0.1$.

Learning curves for the base network using different optimizers are displayed in the center plot of Figure 4. In this experiment NAG, ADAM, and RMSprop exhibit roughly equivalent performance the LSTM optimizer outperforms them by a significant margin. The right plot in Figure 4 compares the performance of the LSTM optimizer if it is allowed to run for 200 steps, despite having been trained to optimize for 100 steps. In this comparison we re-used the LSTM optimizer from the previous experiment, and here we see that the LSTM optimizer continues to outperform the baseline optimizers on this task.

**Generalization to different architectures** Figure 5 shows three examples of applying the LSTM optimizer to train networks with different architectures than the base network on which it was trained. The modifications are (from left to right) (1) an MLP with 40 hidden units instead of 20, (2) a network with two hidden layers instead of one, and (3) a network using ReLU activations instead of sigmoid. In the first two cases the LSTM optimizer generalizes well, and continues to outperform the hand-designed baselines despite operating outside of its training regime. However, changing the activation function to ReLU makes the dynamics of the learning procedure sufficiently different that the learned optimizer is no longer able to generalize. Finally, in Figure 6 we show the results of systematically varying the tested architecture; for the LSTM results we again used the optimizer trained using 1 layer of 20 units and sigmoid non-linearities. Note that in this setting where the

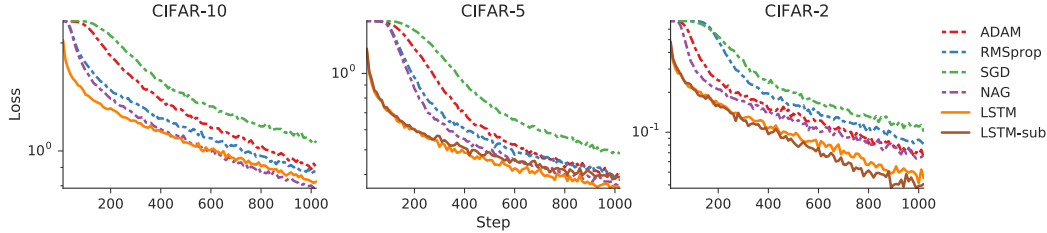

Figure 7: Optimization performance on the CIFAR-10 dataset and subsets. Shown on the left is the LSTM optimizer versus various baselines trained on CIFAR-10 and tested on a held-out test set. The two plots on the right are the performance of these optimizers on subsets of the CIFAR labels. The additional optimizer *LSTM-sub* has been trained only on the heldout labels and is hence transferring to a completely novel dataset.

test-set problems are similar enough to those in the training set we see even better generalization than the baseline optimizers.

### 3.3   Training a convolutional network on CIFAR-10

Next we test the performance of the trained neural optimizers on optimizing classification performance for the CIFAR-10 dataset [Krizhevsky, 2009]. In these experiments we used a model with both convolutional and feed-forward layers. In particular, the model used for these experiments includes three convolutional layers with max pooling followed by a fully-connected layer with 32 hidden units; all non-linearities were ReLU activations with batch normalization.

The coordinatewise network decomposition introduced in Section 2.1—and used in the previous experiment—utilizes a single LSTM architecture with shared weights, but separate hidden states, for each optimizee parameter. We found that this decomposition was not sufficient for the model architecture introduced in this section due to the differences between the fully connected and convolutional layers. Instead we modify the optimizer by introducing two LSTMs: one proposes parameter updates for the fully connected layers and the other updates the convolutional layer parameters. Like the previous LSTM optimizer we still utilize a coordinatewise decomposition with shared weights and individual hidden states, however LSTM weights are now shared only between parameters of the same type (i.e. fully-connected vs. convolutional).

The performance of this trained optimizer compared against the baseline techniques is shown in Figure 7. The left-most plot displays the results of using the optimizer to fit a classifier on a held-out test set. The additional two plots on the right display the performance of the trained optimizer on modified datasets which only contain a subset of the labels, i.e. the CIFAR-2 dataset only contains data corresponding to 2 of the 10 labels. Additionally we include an optimizer *LSTM-sub* which was only trained on the held-out labels.

In all these examples we can see that the LSTM optimizer learns much more quickly than the baseline optimizers, with significant boosts in performance for the CIFAR-5 and especially CIFAR-2 datsets. We also see that the optimizers trained only on a disjoint subset of the data is hardly effected by this difference and transfers well to the additional dataset.

### 3.4   Neural Art

The recent work on artistic style transfer using convolutional networks, or Neural Art [Gatys et al., 2015], gives a natural testbed for our method, since each content and style image pair gives rise to a different optimization problem. Each Neural Art problem starts from a *content image*, $c$, and a *style image*, $s$, and is given by

$$f(\theta) = \alpha \mathcal{L}_{\text{content}}(c, \theta) + \beta \mathcal{L}_{\text{style}}(s, \theta) + \gamma \mathcal{L}_{\text{reg}}(\theta)$$

The minimizer of $f$ is the *styled image*. The first two terms try to match the content and style of the styled image to that of their first argument, and the third term is a regularizer that encourages smoothness in the styled image. Details can be found in [Gatys et al., 2015].

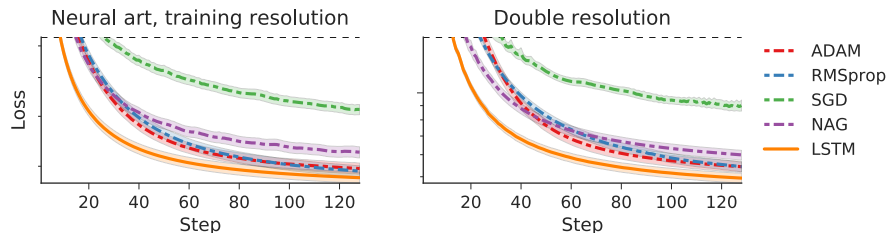

Figure 8: Optimization curves for Neural Art. Content images come from the test set, which was not used during the LSTM optimizer training. Note: the y-axis is in log scale and we zoom in on the interesting portion of this plot. **Left:** Applying the training style at the training resolution. **Right:** Applying the test style at double the training resolution.

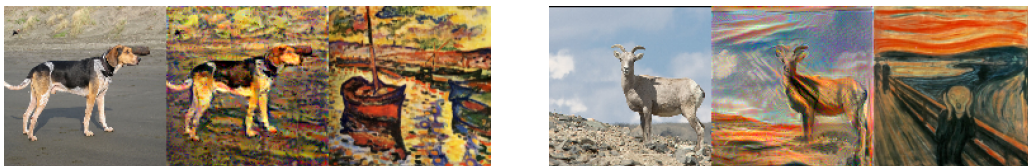

Figure 9: Examples of images styled using the LSTM optimizer. Each triple consists of the content image (left), style (right) and image generated by the LSTM optimizer (center). **Left:** The result of applying the training style at the training resolution to a test image. **Right:** The result of applying a new style to a test image at double the resolution on which the optimizer was trained.

We train optimizers using only 1 style and 1800 content images taken from ImageNet [Deng et al., 2009]. We randomly select 100 content images for testing and 20 content images for validation of trained optimizers. We train the optimizer on 64x64 content images from ImageNet and one fixed style image. We then test how well it generalizes to a different style image and higher resolution (128x128). Each image was optimized for 128 steps and trained optimizers were unrolled for 32 steps. Figure 9 shows the result of styling two different images using the LSTM optimizer. The LSTM optimizer uses inputs preprocessing described in Appendix A and no postprocessing. See Appendix C for additional images.

Figure 8 compares the performance of the LSTM optimizer to standard optimization algorithms. The LSTM optimizer outperforms all standard optimizers if the resolution and style image are the same as the ones on which it was trained. Moreover, it continues to perform very well when both the resolution and style are changed at test time.

Finally, in Appendix B we qualitatively examine the behavior of the step directions generated by the learned optimizer.

## 4   Conclusion

We have shown how to cast the design of optimization algorithms as a learning problem, which enables us to train optimizers that are specialized to particular classes of functions. Our experiments have confirmed that learned neural optimizers compare favorably against state-of-the-art optimization methods used in deep learning. We witnessed a remarkable degree of transfer, with for example the LSTM optimizer trained on 12,288 parameter neural art tasks being able to generalize to tasks with 49,152 parameters, different styles, and different content images all at the same time. We observed similar impressive results when transferring to different architectures in the MNIST task.

The results on the CIFAR image labeling task show that the LSTM optimizers outperform hand-engineered optimizers when transferring to datasets drawn from the same data distribution.

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
