[Supplementary Material]



Figure 10: Updates proposed by different optimizers at different optimization steps for two different coordinates.

# A   Gradient preprocessing

One potential challenge in training optimizers is that different input coordinates (i.e. the gradients w.r.t. different optimizee parameters) can have very different magnitudes. This is indeed the case e.g. when the optimizee is a neural network and different parameters correspond to weights in different layers. This can make training an optimizer difficult, because neural networks naturally disregard small variations in input signals and concentrate on bigger input values.

To this aim we propose to preprocess the optimizer's inputs. One solution would be to give the optimizer $(\log(|\nabla|), \operatorname{sgn}(\nabla))$ as an input, where $\nabla$ is the gradient in the current timestep. This has a problem that $\log(|\nabla|)$ diverges for $\nabla \to 0$. Therefore, we use the following preprocessing formula

$$\nabla^k \to \begin{cases} \left( \frac{\log(|\nabla|)}{p}, \operatorname{sgn}(\nabla) \right) & \text{if } |\nabla| \geq e^{-p} \\ (-1, e^p \nabla) & \text{otherwise} \end{cases}$$

where $p > 0$ is a parameter controlling how small gradients are disregarded (we use $p = 10$ in all our experiments).

We noticed that just rescaling all inputs by an appropriate constant instead also works fine, but the proposed preprocessing seems to be more robust and gives slightly better results on some problems.

# B   Visualizations

Visualizing optimizers is inherently difficult because their proposed updates are functions of the full optimization trajectory. In this section we try to peek into the decisions made by the LSTM optimizer, trained on the neural art task.

**Histories of updates**   We select a single optimizee parameter (one color channel of one pixel in the styled image) and trace the updates proposed to this coordinate by the LSTM optimizer over a single trajectory of optimization. We also record the updates that would have been proposed by both SGD and ADAM if they followed the same trajectory of iterates. Figure 10 shows the trajectory of updates for two different optimizee parameters. From the plots it is clear that the trained optimizer makes bigger updates than SGD and ADAM. It is also visible that it uses some kind of momentum, but its updates are more noisy than those proposed by ADAM which may be interpreted as having a shorter time-scale momentum.

**Proposed update as a function of current gradient**   Another way to visualize the optimizer behavior is to look at the proposed update $g_t$ for a single coordinate as a function of the current gradient evaluation $\nabla_t$. We follow the same procedure as in the previous experiment, and visualize the proposed updates for a few selected time steps.

These results are shown in Figures 11–13. In these plots, the $x$-axis is the current value of the gradient for the chosen coordinate, and the $y$-axis shows the update that each optimizer would propose should the corresponding gradient value be observed. The history of gradient observations is the same for all methods and follows the trajectory of the LSTM optimizer.

The shape of this function for the LSTM optimizer is often step-like, which is also the case for ADAM. Surprisingly the step is sometimes in the opposite direction as for ADAM, i.e. the bigger the gradient, the bigger the update.

## C  Neural Art

Shown below are additional examples of images styled using the LSTM optimizer. Each triple consists of the content image (left), style (right) and image generated by the LSTM optimizer (center).

Figure 11: The proposed update direction for a single coordinate over 32 steps.

Figure 12: The proposed update direction for a single coordinate over 32 steps.

Figure 13: The proposed update direction for a single coordinate over 32 steps.