[Reviews · NeurIPS 2016]

Reviewer 1

Summary

The authors propose a method for learning update functions in gradient-based optimizers. By jointly training an optimizer and optimizee, parameters of an LSTM-based DNN can be learned that accumulate information from multiple gradients, similar to momentum. Once trained, an optimizer can be reused to optimize similar tasks, and can partially generalize to new network architectures. While no theoretical properties of the resulting algorithms are known, several small empirical experiments demonstrate that the resulting optimizers are competitive with state-of-the-art alternatives.

Qualitative Assessment

I really enjoyed this paper - it's creative, timely, interesting, and could have significant impact. I think the idea of "learning algorithms" is an important next step in deep learning; this seems like a reasonable step in that direction. The experimental results seemed fine, but I guess I wish they were done on larger tests. I think everyone would like to know if this idea could help train, say, the next AlexNet. I wish that the post-hoc analysis told us something more about the strategy the optimization nets are learning. It would be interesting to figure out what signal they're capitalizing on, and see if that suggests new theoretical directions for the optimization community. While I was pleasantly surprised to see that the resulting algorithms were better than state-of-the-art alternatives, I was more surprised to see that they weren't better by all that much. I suppose this suggests that our current algorithms are pretty good; a testament to the hard work done by the optimization community. I wish that this demonstrated learning on a problem that simply wasn't solvable before.

Confidence in this Review

3-Expert (read the paper in detail, know the area, quite certain of my opinion)


Reviewer 2

Summary

Paper 1982: Learning to learn by gradient descent by gradient descent An LSTM learns entire (gradient-based) learning algorithms for certain classes of functions, extending similar work of the 1990s and early 2000s.

Qualitative Assessment

Comments (mostly on related work): Authors: "The idea of using learning to learn or meta-learning to acquire knowledge or inductive biases has a long history [Thrun and Pratt, 1998]." But the intro to this reference is muddling the waters by confusing meta-learning (which is about learning the learning algorithm itself) and transfer learning, subsuming basically everything under "meta-learning," even standard back-propagation, because it can be applied to some data set, and then may learn new data points more quickly (so this is just standard transfer learning). To my knowledge, the first work on learning general learning algorithms written in a universal programming language was published in 1987: J. Schmidhuber. Evolutionary principles in self-referential learning, or on learning how to learn: The meta-meta-... hook. Diploma thesis, Institut für Informatik, Technische Universität München, 1987. Authors: "This work was built on by [Younger et al., 2001, Hochreiter et al., 2001] wherein a higher-level network act as a gradient descent procedure, with both levels trained during learning. Earlier work of Runarsson and Jonsson [2000] trains similar feed-forward meta-learning rules using evolutionary strategies. Alternatively Schmidhuber [1992, 1993] considers networks that are able to modify their own behavior and act as an alternative to recurrent networks in meta-learning. Note, however that these earlier works do not directly address the transfer of a learned training procedure to novel problem instances and instead focus on adaptivity in the online setting." This does not quite do justice to the previous work. Schmidhuber's 1993 meta-RNN was not an alternative to RNNs but an RNN able to run arbitrary computable weight change algorithms on the RNN itself. It could sequentially address any of its own weights and read or modify it (a bit like an NTM whose storage cells are actually fast weights). The whole system was differentiable end-to-end such that gradient descent could be used to search for a learning algorithm and also a meta-learning algorithm and a meta-meta-learning algorithm and so on - no restrictions. And Hochreiter's meta-LSTM really did "directly address the transfer of a learned training procedure to novel problem instances" (the words of the authors of the submission) in practice, because it was trained on certain quadratic functions to learn a learning algorithm for quadratic functions, and then a new quadratic function was presented in form of input/target pairs to the input units of the meta-LSTM, which generalized from the previous problem instances (which were functions) to the new ones, such that it could learn the new quadratic function 30 times faster than backpropagation or gradient descent, that is, it really learned by gradient descent to learn much faster than by gradient descent. Whether the resulting learning algorithm used something that's closely related to some fast version of gradient descent (or something more sophisticated) was not analyzed, to my knowledge. But the setup was general enough to permit all kinds of computable learning algorithms. I think it is very important that the authors make very clear what's different to their own approach, provided there is a major difference at all. I think the main difference is that the authors explicitly bias their meta-learning setup towards the learning of gradient-based learning algorithms (while the previous work did not have an explicit bias of this kind). This may be a good thing, at least in the context of the studied experimental tasks, and should be emphasized. Authors: "Finally, Daniel et al. [2016] considers using reinforcement learning to train a controller for selecting step-sizes, however this work is much more constrained than ours and still requires hand-tuned features." However, there was a much earlier much more general reinforcement learning system that learned to learn learning algorithms (written in a general programming language - no restrictions), namely the success-story algorithm SSA (called EIRA in an earlier ICML publication): M. Wiering and J. Schmidhuber. Solving POMDPs using Levin search and EIRA. In L. Saitta, ed., Machine Learning: Proceedings of the 13th International Conference (ICML 1996), pages 534-542, Morgan Kaufmann Publishers, San Francisco, CA, 1996. Based on: J. Schmidhuber. On learning how to learn learning strategies. TR FKI-198-94, TUM, 1994. J. Schmidhuber, J. Zhao, and M. Wiering. Shifting inductive bias with success-story algorithm, adaptive Levin search, and incremental self-improvement. Machine Learning 28:105-130, 1997. This system could run arbitrary algorithms, including arbitrary learning algorithms and meta-learning algorithms affecting its own code, using SSA to favor "good" learning algorithms that optimize reward intake per time interval. It should be easy for the authors of the submission to identify the main differences to their own approach. Authors: "We implement the update rule for each coordinate using a two-layer Long Short Term Memory (LSTM) network" ... Did they use the original LSTM of 1997, or the LSTM with forget gates for the recurrent units which most people use?: F. A. Gers and J. Schmidhuber and F. Cummins. Learning to Forget: Continual Prediction with LSTM. Neural Computation, 12(10):2451--2471, 2000. "an NTM-BFGS optimizer, because its use of external memory is similar to the Neural Turing Machine [Graves et al., 2014]." So it is like the fast weight system of 1992 which separates storage and control? Authors: "Quadratic functions. In this experiment we consider training an optimizer on a simple class of synthetic 10-dimensional quadratic functions." So this is like the experiments with quadratic functions of Hochreiter et al's 2001 paper. Did the authors perform a direct comparison to the 2001 system? Conclusion: Authors: "We have shown how to cast the design of optimization algorithms as a learning problem, which enables us to train optimizers that are specialized to particular classes of functions." Well, this was already shown by the previous work up to 2001 mentioned above. I think the conclusion should focus on what's really new in this paper. It seems to me that the novelty lies in the explicit bias towards gradient-based learning algorithms (while the previous work did not have an explicit bias of this kind). This is probably a good bias for the experiments presented in this paper, and should be emphasized. And the additional experiments (besides those with quadratic functions) are cool, and the results are intriguing! I hope that this paper will help to rekindle interest in gradient-based learning to learn. It should be published, provided the comments above are addressed. I'd like to see the revised version again.

Confidence in this Review

3-Expert (read the paper in detail, know the area, quite certain of my opinion)


Reviewer 3

Summary

This paper proposes a new optimization method to replace traditional methods such as SGD and RMSProp, which are hand-crafted parameters update rules. The proposed method is through an LSTM that learns the optimal parameter update at each step. The proposed method replaced hand-designed update rules with a learned update rule. Evaluated on quadratic functions, MNIST and Neural Art, the proposed LSTM optimizer achieved better performance than traditional optimizer.

Qualitative Assessment

This is a really novel optimizing technique that replace hand-crafted parameters update rules with learnable update rule. The learning curve achieved much better result than traditional methods. It's a well written paper enjoyable to read and very easy to follow. There are a few questions hopefully the authors could address: 1. What is the time cost of running LSTM optimizer? Apparently the LSTM optimizer is not free lunch, the proposed update rule requires much more complicated computation than SGD (computer a LSTM v.s. just do ax+y) how much slower will the training process be? 2. In the result section, the authors provided the loss value over interations, what about the final accuracy? 3. The authors provided experiment results on simple tasks such as quadratic functions, MNIST and Neural Art, how is this method generalizable to modern deep networks such as ResNet on large scale datasets? will it be hard to train?

Confidence in this Review

2-Confident (read it all; understood it all reasonably well)


Reviewer 4

Summary

This paper proposes to simplify the design of the optimizer used to minimize a function by using a meta-optimizer whose input is the gradient of the function to be optimized and whose output is the update to be applied. The idea is definitely interesting and I can see how this would benefit the practitioner. I do think, however, that the paper has difficulty living to its claims and I am afraid that, despite the obvious interest of the idea, there are still many hurdles before this idea or a similar one will be adopted.

Qualitative Assessment

The optimization community is vast and numerous papers are published every year about new optimization techniques. Knowing which technique is the most suited to a particular problem requires lots of experience and, once a particular technique has been chosen, there are usually optimizer hyperparameters to be tuned. Starting from this observation, the paper attempts at providing a meta-algorithm whose output would be optimizers (or, rather, gradient transforms) suited to a wide range of problems. The idea is definitely appealing and, should this method work, be of great significance to the community. The paper in general is well written. The literature review is thorough and the presentation of the various ideas very clear. Though I am not too familiar with LSTMs, I could easily follow the gist of the meta-algorithm as well as the claims made. However, I would like to comment on the actual claims made and how the paper actually delivers on them. First, the paper claims that the proposed architecture could technically model famous algorithms such as L-BFGS. While this is true from a parametric point of view, in that it can model a low-rank linear transformation of the gradient using a history of past updates, it is very far-fetched to claim that "a memory could allow the optimizer to learn algorithms similar to [...] L-BFGS". The presence of "if appropriately designed" does not change the fact that L-BFGS relies on very specific updates and that there is a huge gap between implementing history-based transforms and implementing something similar to L-BFGS. Second, while it is absolutely true that there is a wide variety of optimizers "tailored to specific classes of problems" and that that is indeed a problem, the proposed architecture does nothing to cater to these specific classes of problems. In particular, the LSTM is not adapted to problems involving sparsity (using proximal methods and sparsity-inducing norms), convex problems with finite training sets (tackled with methods such as SAGA or SVRG), strongly-convex problems (L-BFGS) or optimization over a compact set (Frank-Wolfe). In fact, all the comparisons are done with diagonal scalings of SGD. As such, the paper falls short of its claim that "the design of an optimization algorithm can be cast as a learning problem". The conclusion is much more accurate, stating that "learned neural optimizers compare favorably against state-of-the-art optimization methods used in deep learning". Third, the use of an LSTM actually increases the number of parameters the practitioner needs to tune. My lack of knowledge on LSTMs unfortunately prevents me from commenting on the sensitivity of the LSTM to the specific optimizer used but I would have liked an analysis of this issue. Further, unless I am mistaken (and that is entirely possible, I am sure the authors will correct me if that is the case), momentum or other tricks (batch normalization, for instance) were not used for the optimizers the LSTM is compared to. As the difference in performance, albeit significant, seems of the same order as the gain of such tricks, this comparison would greatly benefit the paper. In summary, the idea of designing a model that takes local information about the function as input and outputs an update, using a hidden state to summarize history, is very appealing and could unify a lot of the current techniques using stochastic gradient, or variants thereof. However, while the added complexity of an LSTM would be negligible if the rest lived up to its expectations, it is not the case anymore when the impact is a speedup over a few, well-established methods. Thus, the paper needs to be more thorough about its analysis of the optimization of the LSTM and the impact on a wider range of problems. UPDATE AFTER AUTHORS' FEEDBACK AND DISCUSSION I carefully read the authors' feedback and I think they sidestepped the main issue I was mentioning. I fully agree that designing an optimizer through examples rather than through mathematical properties is appealing but this is not what the paper shows. In particular, it does not show that the work needed to learn that optimizer is smaller than the work needed to tune a first-order optimizer for deep nets. Also, it offers generalization at a very narrow scale, which means that I cannot conclude from reading the paper that the amount of designing work for someone who wishes to solve a broad range of problems is smaller than if they had to tune each optimizer individually. That is why I do not change my score and still think that this paper needs a major rewriting to be fully honest about what it achieves.

Confidence in this Review

2-Confident (read it all; understood it all reasonably well)


Reviewer 5

Summary

The authors investigate the use of recurrent neural networks for learning optimal update strategies in the context of first-order optimization (i.e., they replace hand-crafted update strategies such as steepest descent with the output of recurrent networks). They use one LSTM per coordinate, with two variations that do not obtain significant gains in the experiments with respect to the basic version (using global averaging of some LSTM cells, and a read/write mechanism to an external memory). In three examples, they show that the networks can beat a few selected first-order strategies (e.g. ADAM), and possess a small amount of generalization to problems in similar classes. The benchmarks they use are (i) optimization of unregularized linear least-square; (ii) fitting a neural network to the MNIST dataset; and (iii) solving the optimization problems in the context of the 'artistic style transfer', which is used to fuse together two different images.

Qualitative Assessment

From a conceptual point of view, the idea proposed in the paper is very interesting. It is simple to understand, but it opens for many possibilities, so in my opinion it is a perfect paper for the NIPS conference. Of course, from a practical point of view, the work in its current form is not easy to justify, and in this sense this is the major drawback. First-order optimization algorithms have two main advantages: (i) they can be applied to almost any (sub)differentiable problem; (ii) they are very cheap to implement once the gradients are known. The proposed approach looses both of these advantages: the network must be retrained even for very simple changes in the problem to be optimized (e.g., a change of activation function in the network); training is costly; and the memory/time overhead during the execution of many LSTM networks can be significant. In fact, if one has available all this computing power and time to spend, what is the rationale behind choosing a simple first-order procedure compared to second-order techniques? The method appears to be promising particularly for situations in which there is the need for solving the same optimization problem thousands of times over. In this sense, I guess the most interesting applications can come from outside the realm of machine learning. In fact, the only experiment actually involving the training of a neural network is (in my opinion) the least convincing of the three.

Confidence in this Review

3-Expert (read the paper in detail, know the area, quite certain of my opinion)


Reviewer 6

Summary

In this paper the authors train an LSTM to act as a diagonal* gradient based optimizer. They demonstrate the systems capability to optimize quadratic functions, small MLPs and images for style-transfer faster than mainstream optimizers.

Qualitative Assessment

The topic of the paper is really nice and important, but the paper itself feels rushed. * weight of the individual timesteps: The chosen scheme of putting a weight of 1 to each timestep implies that the system tries to minimize the integral over the error, instead of the final error. This can be an advantage in the beginning since it rewards quick convergence. On the other hand it might lead the system to focus too much on the beginning of the optimization where most of the improvement made. So this issue is at the core of the question "What makes a good optimizer" raised earlier in the paper. It should therefore be discussed more explicitly. * How many optimization runs did it take to train the optimizer? * Optimizers are run for a very short time: Only 1/5 of an epoch on MNIST! What is the reason for this particularly short time? Figure 2(right) gives the impression that the method generalizes to longer horizons, so why not show that? And if it doesn't that would be important to show too! * Discuss limitations! How does the runtime of this model compare to SGD/ADAM? I assume the main limitiation is memory consumption, since during (meta-)training the LSTM optimizer needs to keep about 160 states per parameter per step. * The most important aspect to investigate should be how well the trained optimizers generalize different architectures and datasets! But the paper only investigates three small changes to the architecture, with no error-bars. Having, for example, a systematic investigation about how hidden layer size affects optimizer performance should be easy to do and is painfully missing. * I don't see what the NTM-BFGS optimizer adds to the paper. It is only briefly mentioned and only evaluated on the cubic functions, where it performs the same as the LSTM-GAC. I therefore suggest to remove it. Minor: * in Line 97 it should be be the gradient of f * The sentence "... was optimized for 100 steps and the trained optimizer was unrolled for x steps." occurs several times and is confusing: I assume you mean that *for training* the optimizer was unrolled for x steps. * Line 193: Averaged over how many functions? * Plots are sloppy: No axis labels. Short titles would have helped. * Pre- and Postprocessing are handcrafted and tuned for each problem, which in my opinion collides with the basic tenet of the paper: remove handcrafting from the optimizer. * I would really like to see also other optimizer architectures. How important is LSTM? Two layers? 20 Units?

Confidence in this Review

3-Expert (read the paper in detail, know the area, quite certain of my opinion)